# Spatio-temporal approach to moving window block kriging of satellite data v1.0

Jovan M. Tadić[1], Xuemei Qiu[1], Scot Miller[1] and Anna M. Michalak[1]

[1]{Department of Global Ecology, Carnegie Institution for Science, Stanford, CA 94305, USA}

**Abstract**. Numerous existing satellites observe physical or environmental properties of the Earth system. Many of these satellites provide global-scale observations, but these observations are often sparse and noisy. By contrast, contiguous, global maps are often most useful to the scientific community (i.e., level 3 products). We develop a spatiotemporal moving window block kriging method to create contiguous maps from sparse and/or noisy satellite observations. This approach exhibits several advantages over existing methods: 1) it allows for flexibility in setting the spatial resolution of the level 3 map, 2) it is applicable to observations with variable density, 3) it produces a rigorous uncertainty estimate, 4) it exploits both spatial and temporal correlations in the data, and 5) it facilitates estimation in real time. Moreover, this approach only requires ~~a limited number of assumptions~~ the assumption that the observable quantity exhibits spatial and temporal correlations that are inferable from the data. We test this method by creating Level 3 products from satellite observations of $CO_2$ ($XCO_2$) from the Greenhouse Gases Observing Satellite (GOSAT~~,~~), $CH_4$ ($XCH_4$) from the Infrared Atmospheric Sounding Interferometer (IASI) and solar-induced chlorophyll fluorescence (SIF) from the Global Ozone Monitoring Experiment–2 (GOME-2~~.~~). We evaluate and analyze the difference in performance of spatio-temporal vs. recently developed spatial kriging methods.

## 1. Introduction

Satellite observations of the Earth's surface and atmosphere provide a valuable window into the functioning of the Earth system. Satellites often provide global observations, but these observations are rarely uniform or contiguous in space/time. The observations can be non-contiguous due to satellite orbit geometries and periods, geophysical limitations (e.g. cloud cover), and temporary instrument malfunctions. Furthermore, satellites may provide a large quantity of data, but individual observations can have a large noise-to-signal ratio. It is often necessary to spatially interpolate the data in order to organize the data onto a regular grid, query the data at a particular location of interest, estimate data at unsampled times and/or locations, and/or map the underlying signal in a noisy dataset. These gridded, interpolated maps are commonly named "Level 3" data (e.g. NASA, 2014) and are often part of the standard suite of satellite data products.

$CO_2$ column observations ($XCO_2$) from the Greenhouse Gases Observing Satellite (GOSAT), $CH_4$ column observations ($XCH_4$) from the Infrared Atmospheric Sounding Interferometer (IASI) and solar-induced chlorophyll fluorescence (SIF) observations from The Global Ozone Monitoring Experiment–2 (GOME-2) provide prototypical examples of these challenges, and these three satellites are the primary application used throughout this work (see Section 3).

The most commonly-used method for creating Level 3 maps from satellite data is binning. This approach involves taking the mean of all observations within a given grid cell or "bin" (see Kulawik et al., 2010, and Crévoisier et al., 2009 for examples). The binning method, however, has a number of shortfalls that can lead to inconsistent or inaccurate results. First, different bins contain variable numbers of observations. As a result, some bins will be well-constrained by the data while others may be based upon sparse, noisy

observations. Second, binning does not produce uncertainty estimates. Third, this method cannot
extrapolate the unknown quantity to bins without any observations.
A broad class of geostatistical methods known as kriging provides an alternative approach to mapping
satellite observations. Kriging is a best linear unbiased estimator (for kriging see Chiles and Delfiner, 2012),
where covariance functions are used to represent correlations among data. As a result, kriging can account
for a variable density of observations and can estimate uncertainties in the resulting maps. Various forms
of kriging have recently been used to map satellite Earth observations, particularly for $XCO_2$ (e.g.,
Hammerling et al. 2012a,b; Tadić et al., 2015; Zeng et al., 2013; Guo et al., 2013, Zeng et al., 2016).
Hammerling et al. (2012a,b) presented an approach to map Orbiting Carbon Observatory-2 (OCO-2) and
GOSAT $XCO_2$ observations, respectively, with non-stationary properties. In our previous study (Tadić et
al., 2015) we extended that approach to create $XCO_2$ maps that can have a different spatial resolution from
the resolution or footprint of the original satellite observations. Our previous study and those of
Hammerling et al. (2012a,b) accounted for spatial covariances among observations but did not include a
temporal component. The present study extends this geostatistical framework from a purely spatial to a
spatiotemporal domain.
Spatiotemporal approaches to interpolation can provide a number of advantages relative to purely spatial
methods (e.g. Zeng et al., 2016; Guo et al., 2013). A purely spatial approach will usually aggregate
observations into temporal blocks; observations within the same block effectively have the same time stamp
whether or not those observations are actually synchronous (e.g., Tadić et al., 2015; Hammerling et al.,
2012a,b). Any real temporal variability within a block becomes noise. A spatiotemporal approach, by
contrast, treats time as an explicit dimension and models covariances among data as a function of time. ~~As~~
~~a result, the spatiotemporal approach can (1) fill in temporal gaps in the observations, (2) create maps at~~
~~higher temporal resolutions than purely spatial approach, (3) produce more accurate estimates when~~
~~observations have variable spatio-temporal coverage, (4) predict future values (i.e. extrapolate temporally).~~
A handful of recent studies have considered temporal relationships when mapping satellite observations of
$XCO_2$. These studies have either used various forms of Kalman smoothing (e.g., Katzfuss and Cressie 2011,
Katzfuss and Cressie 2012, Nguyen et al. 2014) or geostatistics (e.g., Guo et al. 2013; Zeng et al. 2013;
Zeng et al. 2016). The former group of studies leverages Kalman smoothing to improve the computational
tractability of mapping dense or abundant datasets, like OCO-2 and the Atmospheric Infrared Sounder
(AIRS). The latter group of studies, by contrast, has applied geostatistics to sparse datasets like those from
the GOSAT satellite. A detailed review of spatial and spatio-temporal mapping methods has been published
recently (Li and Heap, 2014).
The ~~model developed in~~ goal of this ~~paper also uses geostatistics~~study is to ~~map~~develop a geostatistical
spatio-temporal mapping and upscaling method (applicable, but not limited to, satellite observations of
$XCO_2$~~, but we present~~) that exhibits several advances relative to previous ~~efforts.~~methods. It can: (1) fill in
temporal gaps in the observations, (2) create maps at higher temporal resolutions than purely spatial
approach, (3) produce more accurate estimates when observations have variable spatio-temporal coverage,
(4) predict future values (i.e. extrapolate temporally). Among other improvements, we develop an efficient
method ~~to subsample~~for subsampling satellite observations and utilize the product-sum covariance model
(e.g., De Iaco et al., 2001) that is easy to parameterize, which makes it applicable to both ~~abundant~~dense
and sparse datasets. The entire work has been conducted in Matlab 2012a.
Section 2 of this study describes the presented model in detail; it describes an efficient subsampling
procedure that can handle very large datasets and a covariance model that can estimate both spatial and
temporal relationships in the data. We then incorporate these components into a spatiotemporal version of
moving window block kriging. In sections 3 and 4, we subsequently apply this model to map GOSAT
$XCO_2$, IASI $XCH_4$ and GOME-2 SIF at multiple time resolutions (including daily).

## 2. Methods

The spatio-temporal block kriging approach presented in this study proceeds in three steps for each model grid cell and estimation time. First, we subsample the observations within a predetermined spatio-temporal domain (section 2.1). Next, we characterize the local spatio-temporal covariance structure (section 2.2). Finally, we interpolate the satellite observations at the desired spatial resolution (section 2.3).

### 2.1 Subsampling of observations

The ultimate goal of the proposed subsampling strategy is to reduce the number of observations in the spatio-temporal vicinity of an estimation location to a representative, computationally feasible subset of data. We use a subset of observations ($M$) to estimate a local set of covariance parameters and use another subset ($N$) to estimate the desired quantity and associated uncertainty. Note that, for the method presented here, $M$ and $N$ can refer to either the same subset of data or different subsets.

The total number of observations used for covariance parameter estimation ($M$), is selected to be small enough to make this estimation computationally feasible but large enough to yield a sample representative of both local and regional variability. The optimal subset of $N$ observations used for mapping depends on the actually observed covariance structure which is not known prior to covariance parametrization step. In the example presented in Sect. 3, the optimal observational subset used in a mapping step for each grid cell comprised $N$ points having the highest covariance with the estimation location. In the example below, we set both $M$ and $N$ at 500; larger values of $M$ and $N$ did not have a substantial impact on the estimated parameters and mapped quantity, respectively. ~~Furthermore, $M$ should represent local variability, and larger values of $M$ would encompass more distant, non-local regions.~~

We select subset of observations $M$ for each estimation grid cell by assigning a relative selection probability to each observation based on that observation's spatial and temporal 'separation distances' from the centroid of the grid cell. In the absence of a proper metric for distance in space-time, we model the spatial and temporal components of the overall selection probability separately.

The selection probability (and its components) is described by the following equation:

$$P = P_s \times P_t \propto 1/(A_s h_s)^2 \times e^{-(A_t h_t)^2} \tag{1}$$

where $P_s$ is the spatial component of the relative probability of a given observation being selected, $P_t$ is temporal component, $h_s$ and $h_t$ are distances between estimation location and observations, in space and time, respectively, and $A_s$ and $A_t$ are unit dependent, user defined weighting factors between separation distance in space vs. in time (how deep in space vs. time the sampling should occur). The unit dependent choice of $A_s$ and $A_t$ can be initially based on user expectations of the decorrelation distances in space vs. time and, if necessary, subsequently corrected accounting for actually computed decorrelation lengths in space and time in an iterative fashion. In this way temporal and spatial sampling depths could even be locally optimized and become location-specific. In the examples below (Section 3), $A_s$ and $A_t$ were set to 1 km$^{-1}$, and 0.5 day$^{-1}$, respectively, based on the observed average decorrelation distances in space and time (see Fig. 1 and Section 4.1).

[Figure 1]

$h_s$ is calculated as the great circle distance between the centroid $x_j$ of the estimation grid cell and the location $x_i$ of an observation:

$$h_s(x_i, x_j) = r\cos^{-1}(\sin\varphi_i \sin\varphi_j + \cos\varphi_i \cos\varphi_j \cos(\lambda_i - \lambda_j)) \qquad (2)$$
where $\varphi_i$ and $\lambda_i$ are the latitude and longitude of location $x_i$ and $r$ is the radius of the Earth.
The temporal and spatial components of the probability function have different functional forms out of
necessity. The measurements often come pre-aggregated in time slices corresponding to hours, days, or
longer aggregation time periods, which multiplies the number of observations with the same time stamp.
As a result, it is not possible to assign sampling probability along a temporal axis in a manner equivalent to
the spatial approach; doing so would result in infinite probabilities assigned to all observations within the
time slice of the actual estimation location ($P_t \sim 1/0^2 = \infty$). The same holds for spatially co-located
observations. However, since each observation comes with unique spatial coordinates (not pre-binned like
in temporal case), we select a simpler form of the spatial component of the sampling function. The defined
form of $P$ (Eq. 1) ensures that pairs of observations close to estimation location define the shape of the
variogram at short separation distances (the variogram should reflect variability in the spatio-temporal
vicinity of the estimation grid cell. See Section 2.2). Different forms of $P$ can be used if directional
anisotropy is expected or if more/fewer observations along a given direction are desired to better represent
expected correlations.
Previous approaches required the user to choose spatial and temporal windows that determine which
neighboring observations to use (see, for comparison, Alkhaled et al. 2008; Hammerling et al. 2012a,b).
The approach proposed in this paper, by contrast, requires fewer subjective choices – only the form of
sampling function and unit dependent choice of normalizing coefficients $A_s$ and $A_t$. In addition, our
approach is computationally feasible even for very large data sets.

## 2.2 Characterization of Spatio-temporal Covariance

Existing studies have used a number of models to estimate spatio-temporal covariances for a variety of
applications. Models used include the metric model (Dimitrakopoulos and Luo, 1994), linear model
(Rouhani and Hall, 1989), product model (De Cesare et al., 1996), non-separable model (Cressie and
Huang, 1999), and generalized product-sum covariance model (De Iaco et al., 2001). The approach
developed in this paper uses a generalized product-sum covariance model (De Iaco et al., 2001). This model
affords a number of advantages relative to other covariance models: (1) a product sum covariance model
outperformed other models in terms of prediction accuracy in a recent study using GOSAT satellite data
(Guo et al., 2013), (2) it is relatively easy to implement (De Iaco et al., 2001), and (3) it is more flexible
than a non-separable covariance model (De Cesare, 2001a).
The product-sum model, as it has been applied in the past, has one important area for improvement. The
original procedure (De Iaco et al., 2001) assumed separate modeling of the spatial and temporal covariance
(variograms) and their later unification into a spatio-temporal model in the final step. The procedure
requires observations approximately in the same location at multiple different times. However, satellite
observations are often not perfectly collocated in consequent measurement cycles over the same region. As
a result, we would need to assume that each measurement cycle is perfectly co-located with previous/future
cycles, or define an arbitrary tolerance, in order to apply the original approach. This assumption becomes
more prone to error if the observations are very sparse, as is often the case with satellites.
Thus, in this study, we cater to specific properties of satellite data and alter the original procedure by
estimating all covariance parameters simultaneously, thereby avoiding the aforementioned problem.
We broadly define the covariance as follows:
$$C_{s,t}(h_s,h_t) = \mathrm{Cov}(Z(s_+h_s,t_+h_t), Z(s,t)) \tag{3}$$

The equation shows that covariance between two points ($Z$) separated in space-time ($s,t$) depends on their
distance in space ($h_s$) and distance in time ($h_t$). The following class of valid product–sum covariance models
was introduced in De Cesare et al. (2001b) and further developed in De Iaco et al. (2001):
$$C_{s,t}(h_s,h_t) = k_1 C_s(h_s)C_t(h_t) + k_2 C_s(h_s) + k_3 C_t(h_t) \tag{4}$$

where $C_t$ and $C_s$ are valid temporal and spatial covariance models, respectively. De Iaco et al. (2001) proved
that for positive definiteness it is sufficient that $k_1 > 0$, $k_2 \geq 0$ and $k_3 \geq 0$. It is interesting to note that from
Eq. 4 follows that spatio-temporal covariance models collapses down to purely spatial model in cases where
temporal covariance does not exist. Thus, the spatial approach could be viewed as a special case of spatio-
temporal modeling.
The model in Eq. 4 corresponds to the spatio-temporal variogram shown in Equation 5. In the original
procedure, De Iaco et al., 2001 estimated separate spatial ($h_t$=0) and temporal ($h_s$=0) variograms using the
data. De Iaco et al., 2001 ~~then~~ then combined these models to obtain the final spatio-temporal variogram
model:
$$\gamma_{s,t}(h_s,h_t) = \gamma_{s,t}(h_s,0) + \gamma_{s,t}(0,h_t) - k\gamma_{s,t}(h_s,0)\gamma_{s,t}(0,h_t) \tag{5}$$

where $\gamma_{s,t}(h_s,0)$ and $\gamma_{s,t}(0,h_t)$ are spatio-temporal variograms for $h_t$=0 and $h_s$=0, respectively (Figure 2).
Parameter $k$ is estimated from the data which makes the model easily applicable:
$$k = \frac{k_s C_s(0) + k_t C_t(0) - C_{s,t}(0,0)}{k_s C_s(0) k_t C_t(0)} \tag{6}$$

where $k_s C_s(0)$ and $k_t C_t(0)$ are spatial and temporal sills (variances) obtained in modeling of separate
spatial and temporal variograms. The only condition $k$ has to fulfill in order to create an admissible
covariance model is
$$0 < k \leq \frac{1}{max\{\sigma_s{}^2\left(\gamma_{s,t}(h_s,0)\right);\ \sigma_t{}^2\left(\gamma_{s,t}(0,h_t)\right)\}} \tag{7}$$

Due to the specifics of satellite data, we estimate both the covariance parameters and parameter $k$
simultaneously. This approach accounts for constraints that assure a positive definiteness of the model (De
Iaco et al., 2001). This simultaneous approach makes the model more applicable to sparse data and data
with variable spatial coverage, as is often the case with satellite observations.
We use a Gaussian variogram function with a nugget effect to model temporal covariance in the example
presented here (for an overview of variogram models see Chiles and Delfiner, 2012). We use an exponential
model for the spatial variogram. In both cases, we make this choice based upon visual inspection of local
variograms at multiple estimation locations:
$$\gamma_t(h_t)(Gaussian) = \begin{cases} 0, & for\ h_t = 0 \\ \sigma_t{}^2(1 - \exp\left(-\frac{h_t{}^2}{l_t{}^2}\right) + \sigma_{nug}^2, & for\ h_t > 0 \end{cases} \tag{8}$$

$$\gamma_s(h_s)(exponential) = \begin{cases} 0, & for\ h_s = 0 \\ \sigma_s{}^2(1 - \exp\left(-\frac{h_s}{l_s}\right) + \sigma_{nug}^2, & for\ h_s > 0 \end{cases} \tag{9}$$

where $\sigma^2$ and $l$ are the variance and correlation length of the quantity being mapped, and $\sigma^2_{nug}$ is the nugget
variance, typically representative of measurement and retrieval errors in the case of satellite observations.
[Figure 2]
Unlike the original procedure in De Iaco et al. (2001), we model the variogram using only two steps. First,
we calculate a raw spatio-temporal variogram based on the subsampled observations for each estimation
grid cell:

$$\gamma(h_s, h_t) = \tfrac{1}{2}[y(x_i) - y(x_j)]^2 \tag{10}$$

where $\gamma$ is the raw spatio-temporal variogram value for a given pair of observations $y(x_i)$ and $y(\underline{x}_j)$, and $h_s$
and $h_t$ are, respectively, the great circle distance and temporal distance between the spatio-temporal
locations ($x_i$ and $x_j$) of these observations.
Second, we fit the theoretical variogram defined in Eq. 5 to the raw variogram using non-linear least
squares. We subsequently calculate the spatiotemporal covariance using the following equation:

$$C_{s,t}(h_s, h_t) = C_{s,t}(0,0) - \gamma_{s,t}(h_s, h_t)) \tag{11}$$

**Validity on the sphere.** Most covariance models were originally designed for Euclidean space, and their
validity in other coordinate systems cannot be assumed *per se*. Huang et al. (2011) examined the validity
of several theoretical covariance models in spherical coordinate systems. However, this evaluation has not
been done for the spatio-temporal product-sum covariance model. Other studies that use a product-sum
covariance model typically assume the validity of this covariance model on a sphere (e.g., Zeng et al., 2013;
Zeng et al., 2016). Results from Huang et al. (2011) explicitly validate the exponential covariance model
on a sphere, as well as sums of the products of exponential covariance models and constants (provided that
the constants are positive). The first term of the product-sum covariance model used in this study (Eq. 4)
represents a Hadamard product (Million, 2007) of two positive definite matrices. According to Schur
product theorem, a Hadamard product of two positive definite matrices necessarily gives a positive definite
matrix (Mathias, 1993). It therefore follows that a generalized product-sum model (Equation 4) is valid on
a sphere if its spatial component is valid on a sphere.
**2.3 Mapping using spatio-temporal moving window block kriging**
This section leverages the sampling function (Sect. 2.1) and the product-sum covariance model (Sect. 2.2)
to implement a spatio-temporal version of moving window block kriging. A primary advantage of block
kriging is its ability to estimate contiguous maps at any spatial resolution equal to or coarser than the spatial
support (i.e. footprint size) of observations (refer to Sect. 1 and Tadić et. al. 2015). Unlike ordinary kriging
method, the spatial support in block kriging corresponds to the average value within each chosen grid cell.
Moving window block kriging requires solving a set of linear equations to obtain a set of weights ($\lambda$). These
weights must be estimated for each prediction location using N associated observations:

$$\begin{bmatrix} \mathbf{Q} + \mathbf{R} & \mathbf{1} \\ \mathbf{1}^T & 0 \end{bmatrix} \begin{bmatrix} \lambda \\ -\nu \end{bmatrix} = \begin{bmatrix} \mathbf{q}_A \\ 1 \end{bmatrix} \tag{12}$$

In this equation, $\mathbf{R}$ is a diagonal $N{\times}N$ nugget covariance matrix that describes measurement and retrieval
errors, $\mathbf{Q}$ is a $N{\times}N$ covariance matrix among the $N$ observations with individual entries as defined in Eqn.
11, $\mathbf{1}$ is an $N{\times}1$ unity vector, $\nu$ is a Lagrange multiplier, and $\mathbf{q}_A$ is an $N{\times}1$ vector of the spatio-temporal
covariances between the $N$ observation locations and the estimation grid cell, defined as:
$$q_{A,i} = \frac{1}{n}\sum_{j=1}^{n} q\left(h_{s_{i,j}}, h_{t_{i,j}}\right) \tag{13}$$

where $q_{A,i}$ is the covariance between the grid cell and observation $i$. $q(h_{i,j})$ is defined as $C_{s,t}$ in Eqn. 11
based on the distances $h_{s_{i,j}}$ and $h_{t_{i,j}}$ between observation $i$ and $n$ regularly-spaced locations within the grid
cell. In the context of satellite measurements, $n$ is a highest number of non-overlapping footprints contained
within a grid cell and was calculated based on the relative size of the satellite footprint compared to the
size of the estimation grid cells. $n$ varies with latitude, as the size of grid cells decreases with the distance
from the equator. The system in Eqn. 12 is solved for the weights ($\lambda$) and the Lagrange multiplier ($v$). We
subsequently use these parameters to define the estimate ($\hat{z}$) and estimation uncertainty ($\sigma^2_{\hat{z}}$) for the grid
cell:
$$\hat{z} = \lambda^T \mathbf{y} \tag{14}$$

$$\sigma^2_{\hat{z}} = \sigma_{AA} - \lambda^T \mathbf{q}_A + v \tag{15}$$

where $\mathbf{y}$ is the $N \times 1$ vector of subsampled observations, and $\sigma_{AA}$ is the variance of the observations at the
resolution of the estimation grid cell, defined as:
$$\sigma_{AA} = \frac{1}{n^2}\sum_{j=1}^{n}\sum_{k=1}^{n} q\left(h_{j,k}\right) \tag{16}$$

In that equation, $q\left(h_{s_{i,j}}, h_{t_{i,j}}\right)$ is defined as $C_{s,t}$ in Eqn. 11 based on the distances $h_{s_{i,j}}$ and $h_{t_{i,j}}$ between
any combination of the $n$ regularly spaced locations within the grid cell defined previously.
# 3. Example applications
We select three case studies of satellite Level 2 data to demonstrate the properties of the method developed
in this paper: column-integrated dry air model fraction of $CO_2$ ($XCO_2$) from the Japanese Greenhouse Gas
Observing SATellite (GOSAT), $CH_4$ ($XCH_4$) from the Infrared Atmospheric Sounding Interferometer
(IASI), and solar-induced fluorescence (SIF) the Global Ozone Monitoring Experiment–2 (GOME-2).
Level 2 datasets from GOSAT, IASI and GOME-2 have relatively different characteristics. For example,
GOSAT observations are sparse while IASI and GOME-2 are abundant. These diverse datasets are therefore
ideal for testing the method developed here.
The method was demonstrated by producing two different sets of maps. First, it was applied at resolutions
coarser than native ($1 \times 1°$, $2.5 \times 2°$, and $1 \times 1°$ for GOSAT, IASI and GOME-2, respectively) to
demonstrate block kriging capabilities of the method (Section 3). Second, it was applied at the native
resolution of the satellites for cross-validation (method evaluation) purposes (Section 4).
## 3.1 Total column $CO_2$ ($XCO_2$) observed by GOSAT
The Japanese Greenhouse Gas Observing SATellite (GOSAT) (e.g., Kuze et al., 2009), the first satellite
dedicated to global greenhouse gas monitoring, was launched in 2009. Basic information about the satellite,
its orbit configuration, and the $CO_2$ column observations are given in our previous study (Tadić et al., 2014).
It flies in a polar, sun-synchronous orbit with a 3-day repeat cycle and an approximate 13:00 LT overpass
time. GOSAT has a nadir footprint of about 10.5 km diameter at sea level (Kuze et al., 2009) and $2 \times 10^3$

observations per week. The $XCO_2$ observations from GOSAT have large retrieval uncertainties (e.g., O'Dell et al. 2012) and exhibit large spatial and temporal gaps (e.g., Fig. 3a). Although these $XCO_2$ observations are sparse and noisy, contiguous Level 3 maps are often desirable for environmental and ecological applications (Maksyutov et al., 2013; Liu et al., 2012). To this end, we generate global daily estimates for $XCO_2$ (August 2-7, 2009) to match the timeframe used in Tadić et al., 2014.

[Figure 3]

We obtain bias-corrected and filtered GOSAT Level 2 observations using NASA's Atmospheric $CO_2$ Observations from Space (ACOS) algorithm v3.4 release 3 (e.g., O'Dell et al., 2012; Crisp et al., 2012). In this study, we use spatio-temporal moving window block kriging to create a series of contiguous, in-filled global daily maps and associated uncertainties for 2-7 August 2009 (two repeat cycles) (Fig. 3a-c) at $1\times1^o$ resolution. We select the time period to match the time period from our previous study (Tadić et al., 2014). Unlike results from our previous study and other similar studies, which created estimates at 6-day or longer time periods (Hammerling et al., 2012a), we leverage the method developed here to produce maps at the daily scale.

## 3.2 Total column $CH_4$ ($XCH_4$) observed by IASI

The Infrared Atmospheric Sounding Interferometer (IASI) developed by the Centre National d'Etudes Spatiales (CNES) in collaboration with the European Organisation for the Exploitation of Meteorological Satellites (EUMETSAT) is a Fourier Transform Spectrometer based on a Michelson Interferometer coupled to an integrated imaging system that measures infrared radiation emitted from the Earth. It is carried by MetOp-A, a sun-synchronous polar orbit satellite which flows at an altitude of 817 km. Detailed information about the IASI instrument could be found elsewhere (Crévoisier et al., 2009a,b; Massart et al., 2014). IASI has an instantaneous field of view of $50\times50$ km, composed of four pixels each 12 km in radius, delivering $\sim56\times10^3$ $XCH_4$ observations per week.

[Figure 4]

Methane Level 2 IASI (0-4 km) data were retrieved at the NOAA/NESDIS using the NUCAPS (NOAA Unique CrIS/ATMS Processing System) algorithm (Gambacorta, 2013; Xiong et al., 2013). For the ice-covered ocean the data for the lower troposphere (0-4 km) are unreliable due to insufficient thermal contrast between the surface and the atmosphere. Filtering parameters have been provided by Xiong (2014, private communication). The data are available at http://www.nsof.class.noaa.gov/. Using the new method, we created a series of contiguous global daily maps and associated uncertainties for the Northern Hemisphere, for February 26-March 4, 2013 (i.e. Figure 4a-c) at $1^o\times1^o$ resolution. We chose this time period to match the occurrence of the methane "anomaly" North of the coast of Scandinavia.

## 3.3 Global land solar-induced fluorescence fields observed by GOME-2

The GOME-2 (The Global Ozone Monitoring Experiment–2) instrument on board METOP-A (e.g., Joiner et al., 2013) observes solar-induced fluorescence (SIF). The GOME-2 spatial footprint (i.e. support) of the observations is 40 km $\times$ 80 km (Joiner et al, 2013), and the volume of available data is approximately $2\times10^5$ SIF observations per week.

[Figure 5]
Multiple recent studies have demonstrated the potential use of satellite observations of solar-induced
fluorescence (SIF) for understanding the photosynthetic $CO_2$ uptake at large scales (Joiner et al., 2011;
Joiner et al., 2012; Joiner et al., 2013; Frankenberg et al., 2011; Frankenberg et al., 2012; Guanter et al.,
2012, Lee et al., 2013; Frankenberg et al., 2014). Satellite SIF measurements can be used with land surface
models to understand GPP response to environmental stress (e.g., Lee et al., 2013) and to improve the
representation of GPP. GOME-2 provides the highest spatial and temporal density of data, among all
available datasets.
In the example presented here we use SIF GOME-2 v.14 data (Joiner et al., 2013) with the approach
described in Section 2 to create contiguous maps of SIF at a single spatial resolution (1º × 1º) and daily
temporal resolutions. Maps of SIF and associated uncertainties are created at daily temporal resolutions
covering 5-14 May, 2012, some of which are shown on Figures 5a-c.

# 321    4. Method evaluation: accuracy, precision and bias

## 322    4.1 Accuracy, precision and bias

We use a leave-one-out cross validation technique to assess the performance of spatio-temporal (ST) versus
spatial moving window block kriging. We produce these estimates at the native resolution of GOSAT, IASI
and GOME-2 satellites/instruments, which allowed a direct comparison to measured values. For IASI and
GOME-2, for each day in February 26-March 4, 2013, and May 5-14, 2012, respectively, 10% of available
observational data were randomly selected for use in leave-one-out cross-validation and their coordinates
extracted. For $XCO_2$, all GOSAT $XCO_2$ observations for each day in August 2-7, 2009, were used. We
assess the accuracy (the difference between estimates and withheld observations) of both methods using
two common measures: (1) Mean Absolute Error (MAE), and (2) Root Mean Squared Error (RMSE). We
alsoWe also use two more recently proposed measures (Li and Heap, 2011; Li, 2016) that remove the effect
of unit/scal. The first is relative mean absolute error (RMAE) that is given as:
$$RMAE = \frac{1}{n}\sum_{i=1}^{n}|(\hat{z}_i - y_i)/o_i| \times 100 \hspace{3cm} (17)$$
and the second is relative root mean square error (RRMSE), as follows:
$$RRMSE = \left[\frac{1}{n}\sum_{i=1}^{n}(|y_i - \hat{z}_i|/y_i)^2\right]^{1/2} \times 100 \hspace{2cm} (18)$$
where $n$ is the number of observations or samples, $o$ is observed value, and $p$ is predicted or estimated value.
We assess the performance of each method using two additional measures: (3) the accuracy of the
uncertainty bounds (the degree to which the reported uncertainties capture the difference between estimates
and withheld observations) and (4) bias (the mean difference between estimates and withheld observations).
We parameterize the temporal component of the spatio-temporal sampling function in such a way
that observations located +/- 3 days from the actual date had 10% probability of being sampled

compared to observations from the actual day (see Fig 1a). We compare the results to spatial kriging estimates obtained in two different ways, based on observations only from the actual day (1d) and based on observations from +/-3 days from the actual day (7d). This latter case is analogous to the +/- 3-day window that we use for the ST approach. In this 7d case, we obtain these spatial kriging results by assuming the entire observational dataset collected within the selected time period (actual day +/- 3 days) is perfectly temporally correlated. In other words, we use all observations as though they were collected at the same time. We then produce estimates at locations of observations collected within the selected timeframe and compare the performance of the two methods. We repeat procedure described in Section 2 for every observation selected for cross-validation, and we average the statistics, displayed in Table 1.

[Table 1]

According to the results, the spatio-temporal approach performs better than the spatial (7d) approach in all three cases and in all performance measures (for example, spatial (7d) MAE was 6-10% larger). The comparison clearly shows that proper characterization of the temporal covariance between two points residing in different time periods (days), embedded into spatio-temporal approach, improves kriging performance. In IASI ~~case~~and GOME-2 cases, the spatio-temporal method also performed better than spatial (1d). However, in case of GOSAT ~~and GOME 2~~ data, spatio-temporal approach slightly underperformed the spatial (1d) approach having 12% higher MAE (please see Section 4.2 for discussion).

We observed that RMAE and RRMSE error measures should be used with caution in cases when observations can take real zero values, like in the GOME-2 case. In such cases the division by close-to-zero values result in extremely high RMAE and RRMSE values, which overall limits the applicability of these error measures.

We evaluate the accuracy of the uncertainty bounds by examining how often those bounds encapsulate withheld observations. The percentage of observations that fall outside the uncertainty bounds in spatio-temporal approach is comparable to that of the spatial method, confirming the accuracy of the estimated uncertainty bounds (for normally-distributed data the percentage of observations that fall outside of the one, two, and three ~~estimation~~estimations standard deviation ($\sigma_\hat{z}$) uncertainty bounds should be 32%, 5% and 0.3%, respectively). The fraction of observations that fall outside the uncertainty bound is generally lower than would be expected for normally-distributed data, and our results may indicate non-normal features in the data.

## 4.2 When is spatio-temporal modeling recommended?

A ST approach can afford advantages over purely spatial methods when temporal data correlations and data coverage are strong. Indeed, in many cases, the ST approach is more accurate than a purely spatial method (Table 1). This result is consistent with existing literature which uniformly reports that ST approaches are more accurate than spatial approaches (Zeng et al., 2013; Guo et al., 2013; Zeng et al., 2016).

However, although considering information from days preceding and following the target estimation day should in principle always provide a further constraint on the estimate, this does not guarantee that an ST method will always outperform a spatial-only method in practice. The prime reasons for this are two-fold. First, because computational limitations cap the number of observations that can be considered, considering observations across multiple days necessarily leads to a reduction in the spatial density of observations

being considered. This first factor can be partially alleviated by carefully designing the selection probability
function (Eqn. 1). The second reason is that implementing a ST approach involves the estimation of a larger
number of covariance parameters (Eqn. 4-9) relative to a spatial-only approach, which can introduce
additional uncertainty. Indeed, we observe that the purely spatial approach performs better than the ST
method in some cases (e.g., the GOSAT ~~and GOME-2 1d cases).~~case).
Overall, a ST approach is likely to outperform a spatial-only approach when the data exhibit one (or more)
of three characteristics. First, a ST approach is likely better when the data are sparse or unequally
distributed. In these cases, a ST approach can intelligently leverage data in adjacent time periods to
compensate for the sparsity of data in the time period of interest. Second, an ST approach works well for
datasets with temporal gaps (e.g., due to cloud cover or instrument malfunction). An ST approach can fill
these gaps while a spatial-only approach cannot be used for temporal gap-filling. Third, an ST-approach is
well-suited to datasets with regional biases that manifest in one time slice but that do not repeat in adjacent
time slices. ~~Phrased differently, an ST-approach is well-suited to datasets with~~The difference between the
performance of ST and S-approaches obtained through cross-validation becomes most pronounced in
processing datasets with measurement errors that are spatially but not temporally correlated. In these cases,
an ST approach can use data from adjacent time periods to ~~create the~~obtain an estimate, data that do not
have the same regional, spatially-correlated biases. Although the resulting estimate may appear inferior
during cross-validation, this is because that estimate will not reproduce regional biases in data from the
time slice of interest. A spatial-only approach, by contrast, will reproduce these regional biases because it
does not use data from adjacent times when creating the estimate. As a result, a spatial-only approach will
appear to perform better in cross validation, but the ST approach will more accurately reflect the true,
underlying process.

## 407 5. Conclusions

In this study, we develop a method to create high spatio-temporal resolution maps from satellite data using
spatio-temporal moving window block kriging based on product-sum covariance model. The method relies
on a limited number of assumptions: that the observed physical quantity is spatio-temporally auto-
correlated, and that its nature can be inferred from the observations.
The method has several advantages over previously applied methods~~, as alluded to in Sect. 1: 1) it allows~~
~~for the creation of contiguous maps at varying spatio-temporal resolution, 2) it can create maps at temporal~~
~~resolutions shorter than achievable by other binning or kriging methods, 3) it can be applied for creating~~
~~contiguous maps for physical quantities with varying spatio-temporal coverage (i.e., density of~~
~~measurements), 4) it provides assessments of the uncertainty of interpolated values, 5) it utilizes all spatio-~~
~~temporally available information to generate estimates, 6~~. Apart from the advances alluded to in Sect. 1: 1)
it improves covariance parameters estimation procedure because it does not model spatial and temporal
covariance separately, ~~7~~2) it allows for great flexibility in the choice of sampling function and ~~8~~3) it
provides estimates even for the time periods when measurements are not available. It can exploit
correlations with both past and future periods of the observed time spot to provide the most accurate
estimates.
We demonstrate the applicability of this method by creating Level 3 products from the GOSAT $XCO_2$, IASI
$CH_4$ and GOME-2 SIF data. Sparse $XCO_2$ observations from GOSAT and dense $XCH_4$ and SIF
observations from IASI and GOME-2 make a perfect test ground for the method. We show that the proposed
method can even map $XCO_2$ on daily time scales. The method generally yields more precise and accurate
(and unbiased) estimates compared to spatial method which used the same observations but assumed perfect
temporal correlation between data. The factors which could affect the performance of the ST method are
discussed in Section 4.2.
This approach could be used in the future to produce real-time estimates not only of $XCO_2$, $XCH_4$ or SIF,
but of other environmental data observed by satellites which exhibit spatio-temporal autocorrelations.
Especially important could be satellite datasets that have spatially, but not temporally, correlated errors. In
such cases, sampling across several time periods could perhaps help isolate and remove them, which should
be a subject of further studies.
The method could be applied in a standalone mode or as part of a broader satellite data processing package.
Maps produced by the spatio-temporal approach could then be incorporated into physical and
biogeochemical models of the Earth system.
## 6. Code availability
The documented Matlab source code is available at the Researchgate website
(https://www.researchgate.net/publication/311595272_Spatio-temporal_approach_to_moving_window_
_block_kriging_of_satellite_data_v10_code; DOI: 10.13140/RG.2.2.21411.04643). The code is made
available under 'CC BY' license terms (https://creativecommons.org/licenses/).
**Acknowledgement**
This work was supported by the National Aeronautics and Space Administration (NASA) through grant no.
NNX12AB90G and NNX13AC48G, and the National Science Foundation (NSF) through grant no.
1342076. Satellite $CH_4$ IASI v5 data are supplied by the NOAA National Environmental Satellite, Data,
and Information Service (NESDIS): http://www.nsof.class.noaa.gov/. We would also like to thank Leonid
Yurganov (JCET) and Nathaniel Lebedda (University of Maryland) for helpful information and discussions.

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

**Table 1**. Cross-validation results of GOSAT XCO$_2$, IASI XCH$_4$ and GOME-2 SIF datasets using spatio-
temporal and spatial methods, including mean absolute error (MAE), root mean squared error (RMSE),
relative mean absolute error (RMAE), relative root mean square error (RRMSE), percent of observations
lying outside of one, two, and three standard deviations ($\sigma_{\hat{z}}$) of the mapping uncertainty, and mean
difference. MAE, RMSE and bias units for GOSAT, IASI and GOME-2 are ppm, ppb and mW/m$^2$/sr/nm,
respectively. RMAE and RRMSE are unitless, and due to the reasons explained in Section 4.1 given only
for GOSAT and IASI. Shaded fields represent best estimate in each category for every satellite.

| | | GOSAT XCO$_2$ | | | IASI XCH$_4$ | | | GOME-2 SIF | | |
|---|---|---|---|---|---|---|---|---|---|---|
| | | ST | 1d | 7d | ST | 1d | 7d | ST | 1d | 7d |
| **Estimates** | Mean absolute error (MAE) | 0.83 | 0.74 | 0.8889 | 19.19 | 20.23 | 21.0304 | 0.52 | 0.5154 | 0.6654 |
| | Root mean squared error (RMSE) | 1.12 | 0.98 | 1.21 | 25.25 | 27.10 | 27.77 | 0.6768 | 0.6569 | 0.8769 |
| | Relative mean absolute error (RMAE) | 0.22 | 0.19 | 0.23 | 1.04 | 1.09 | 1.14 | - | - | - |
| | Relative root mean square error (RRMSE) | 0.29 | 0.25 | 0.31 | 1.37 | 1.46 | 1.50 | - | - | - |
| **Uncertainties** | % observations falling outside 1$\sigma_{\hat{z}}$ uncertainty | 9.13 | 15.03 | 10.70 | 11.02 | 9.06 | 13.84 | 14.60 | 12.14 | 24.80 |
| | % observations falling outside 2$\sigma_{\hat{z}}$ uncertainty | 1.12 | 3.01 | 1.39 | 0.48 | 0.51 | 0.86 | 1.20 | 0.64 | 4.33 |
| | % observations falling outside 3$\sigma_{\hat{z}}$ uncertainty | 0.067 | 0.52 | 0.13 | 0.04 | 0.046 | 0.022 | 0.11 | 0.05 | 0.83 |
| **Bias** | Mean difference | -0.012 | 0.0066 | -0.034 | 0.28 | -0.14 | 0.58 | 0.016 | 0.0013 | 0.032 |




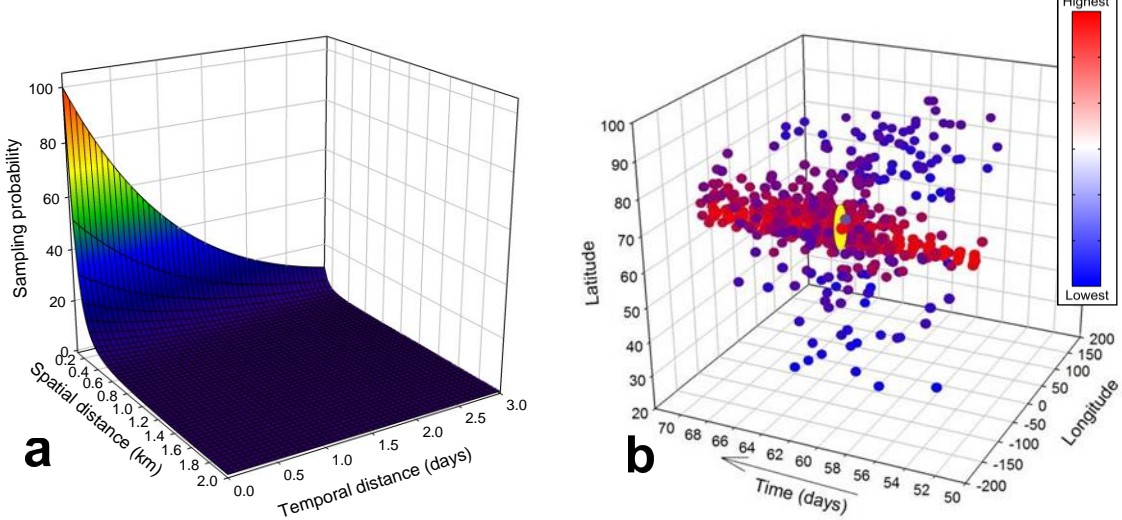


**Figure 1**. (a) Sampling probability as a decreasing function of spatial and temporal distance as used in this
study, (b) The typical example of subsampled IASI Level 2 $XCH_4$ (altitude below 4 km) data for a selected
estimation location (yellow circle). Color of observations shows semivariance between observation and
estimation location (blue-lowest, red-highest). Due to stronger temporal covariance, the relative decrease
of the sampling probability along temporal axis is smaller than with spatial distance.


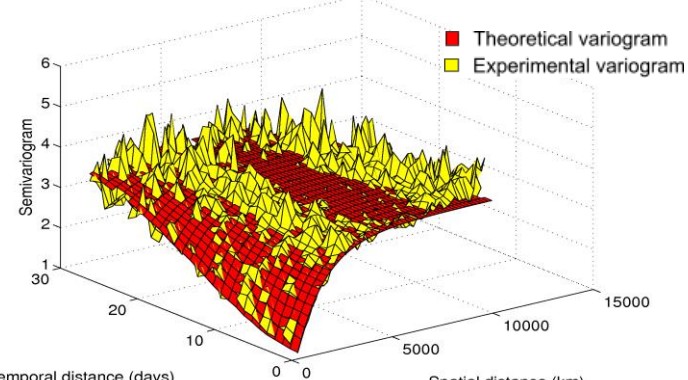


**Figure 2.** Illustration of experimental and fitted theoretical spatio-temporal variogram for GOSAT $XCO_2$
data.

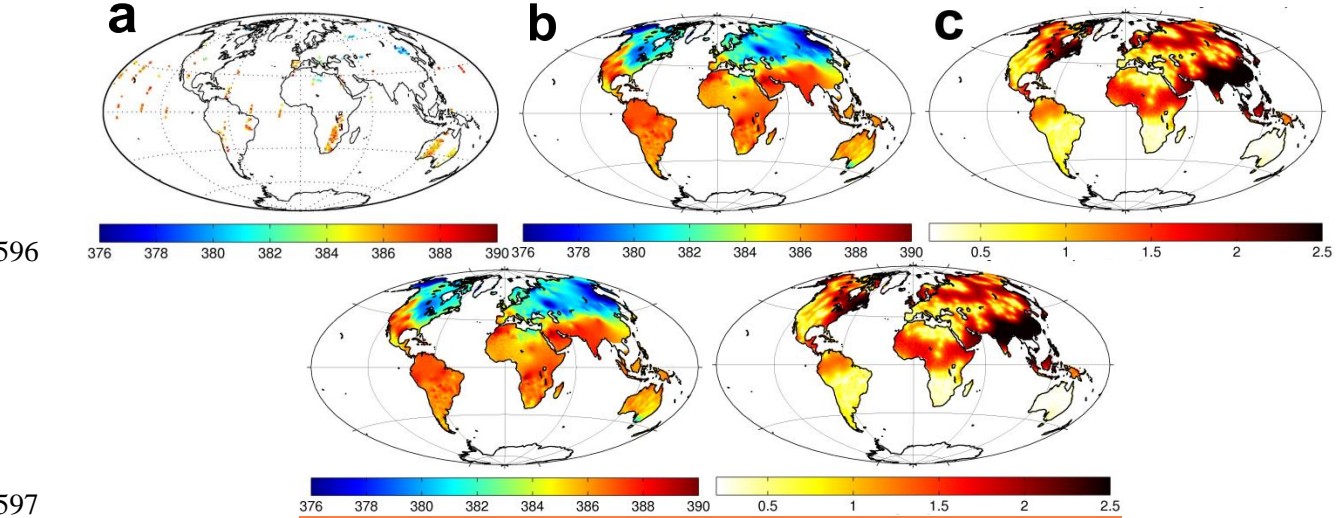



**Figure 3.** (a) GOSAT/ACOS v3.4 XCO$_2$ retrievals (Level 2 data) (ppm) for August 3, 2009 (b) Contiguous global GOSAT/ACOS v3.4 maps (Level 3 data) (ppm) for the same day obtained using Spatio-temporal Moving Window Block Kriging at $1 \times 1°$ spatial resolution, (c) associated uncertainties, given as 1-sigma ($\sigma_{\hat{z}}$) (ppm).



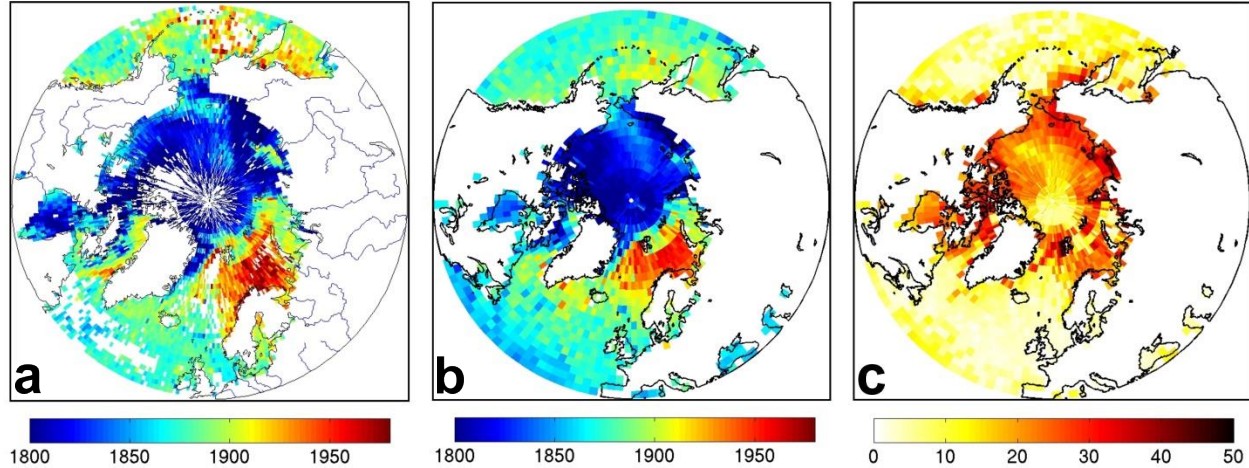


**Figure 4.** (a) IASI XCH$_4$ (0-4 km) retrievals (ppb) for March 2, 2013 (sea only), (b) Contiguous IASI maps for Northern Hemisphere for the same day obtained using Spatio-temporal Moving Window Block Kriging at $2.5 \times 2°$ spatial resolution and (c) associated uncertainties, given as 1-sigma ($\sigma_{\hat{z}}$) (ppb).



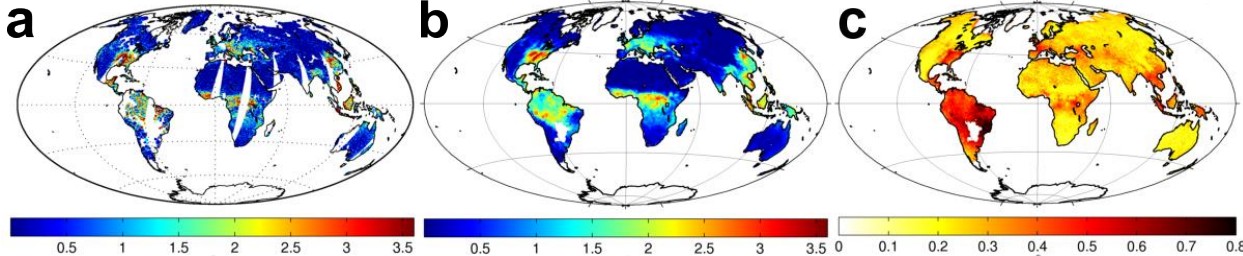

Figure 5. (a) GOME-2 SIF v14 retrievals (Level 2 data) (mW/m$^2$/sr/nm) for May 5, 2012, (b) Contiguous global GOME-2/SIF v14 maps (Level 3 data) (mW/m$^2$/sr/nm) for the same day obtained using Spatio-temporal Moving Window Block Kriging at $1 \times 1°$ spatial resolution, (c) associated uncertainties, given as 1-sigma ($\sigma_{\hat{z}}$) (mW/m$^2$/sr/nm).

