# Peer review of "Spatio-temporal approach to moving window block kriging of"

_Geoscientific Model Development, 2016_

## Referee Comment (RC1) · Anonymous Referee #1 · 17 Oct 2016

Manuscript Number: gmd-2016-192 Title: Spatio-temporal approach to moving window block kriging of satellite data Author(s): J. M. Tadić, X. Qiu, S. Miller, and A. M. Michalak.

This paper intended to introduce an improved spatial-temporal approach to moving window block kriging of satellite data. It provided technique details of the method and compared its performance with spatial-only methods based on three datasets. The accuracy was validated using leave-one-out cross-validation. It is a well prepared and articulated paper. However, some minor issues need to be addressed before publication. I also provide a couple of references at the end of the comments for the authors to further improve the paper.

1. The aims of this study are scattered in the introduction and should be clearly presented in the end of the introduction. 2. As to spatial-temporal approach, it seems some recent developments since 2011 have been missed, which should be included. Please see reference 1 for details. 3. Samples size is missing for the three datasets. Please provide. 4. It is not clear what software was used for this study. Please refer your readers to it so that they could apply your method to their study. 5. The accuracy measures used, MAE and RMSE, are data unit/scale and variation dependent as detailed in reference 2. Please see the recommendations in this reference for accuracy measure selection. 6. A statistical test of the cross-validation results in Table 1 may provide more convincing evidence to show the difference between the methods compared. 7. The conclusion: it is largely repeating what has been presented in the previous sections. It could be condensed by removing the repetitions.

Some further minor issues: 1. Spell out GOSAT, IASI and GOME-2 in the abstract or delete them. 2. Lines 98-99: this sentence suggests that the method is only applicable for a small region. Please revise and clarify. 3. Lines 144-145: are 'generalized product-sum model' and 'generalized product-sum covariance model' the same? Please keep the name consistent in the paper. 4. Line 173: delete one 'then'. 5. For XCO2, only 6 days data were used. Is this too short for ST method? Is it a factor for the poor performance of ST method? 6. Lines 345-346: ST method seems not that poor for GOME-2 data. Please revise.

Refs: 1. J. Li, A. D. Heap, Spatial interpolation methods applied in the environmental sciences: A review. Environmental Modelling & Software 53, 173 (2014). 2. J. Li, Assessing spatial predictive models in the environmental sciences: accuracy measures, data variation and variance explained. Environmental Modelling & Software 80, 1 (2016).

---

## Short Comment (SC1) · 24 Oct 2016

Dear authors,

In my role as Executive editor of GMD, I would like to bring to your attention our Editorial version 1.1:

http://www.geosci-model-dev.net/8/3487/2015/gmd-8-3487-2015.html

This highlights some requirements of papers published in GMD, which is also available on the GMD website in the 'Manuscript Types' section:

http://www.geoscientific-model-development.net/submission/manuscript_types.html

In particular, please note that for your paper, the following requirements have not been met in the Discussions paper:

[Figure]

- "The main paper must give the model name and version number (or other unique identifier) in the title."

- Inclusion of Code and/or data availability sections is mandatory for all papers and should be located at the end of the article, after the conclusions, and before any appendices or acknowledgments. For more details refer to the code and data policy.

Please correct this in your revised submission to GMD.

Yours,

Astrid Kerkweg
* * *

---

## Referee Comment (RC2) · Anonymous Referee #2 · 23 Nov 2016

Review of Tadic et al. "Spatio-temporal approach to moving window block kriging of satellite data"

This paper introduces a spatio-temporal model of environmental data that can be used to produced maps of more uniform coverage, than are capable from the raw satellite observations alone. It applies this approach to three environmental variables from the carbon cycle science community: column co2 (XCO2), column methane (XCH4), and solar-induced chlorophyll fluorescence (SIF). This work follows on a long series of papers by a number of authors with similar models.

**General Comments**

I will be upfront and say that as as reviewer, I am not well-versed in the geostatistical estimation literature, and am rather an expert on these carbon cycle variables themselves. So my review will focus less on the details of this particular approach, and rather some bigger picture questions.

My main complaint on this work, which honestly is more a complaint about the entire field who does this, and is not particular to this paper, is that it fails to really explain the utility of kriged satellite data beyond simply "pretty pictures". Most data users who attempt to extract scientific results from the data do not use 3D maps. The reason is the data assimilation systems typically ingest the sounding (level-2) data directly (e.g., Houweling et al., 2016; Massart et al., 2016). Therefore, some commentary (like a paragraph in the introduction section) on the use of level-3 maps vs. direct data assimilation approaches would be worthwhile, perhaps pointing to scientific results using this method that would have been missed otherwise.

Beyond that, the few basic statistics on the quality of the spatio-temporal (ST) method over and above pure spatial methods do not really argue that the ST approach buys you much. The actual statistics given in Table 1 are really rather similar between pure spatial vs. the ST method. So the paper seems to argue that this is really useful, but the data really don't back it up. My read is that 1-3 day spatial approaches are really quite adequate for this purpose.

Finally, the validation approach is probably not valid for the GOSAT case. This is because there are only ~14 orbits per day, and huge swaths of the globe are missing even if all the data are used. Therefore, you don't really learn the error statistics unless you perform a simulation-based test where you start with a "true" map, sample it like the satellite would, along with realistic observation errors, and then run it through the kriging algorithm to reconstruct the 1-day map. This paper would be much enhanced if such a realistic validation test were performed. I realize the authors can easily say "beyond the scope of this paper" because what I am suggesting is not easy, but it is really the only way I can see to get at the true errors in the proposed algorithm.

**Specific Comments**

Abstract: Makes that statement that this approach only requires a limited number of assumptions – that "the observable quantity exhibits spatial and temporal correlations that are inferable from the data." But this seems like a single assumption? Are there more assumptions? Please reword as necessary.

Section 2.1 I don't get why subsampling is necessary. The data volumes don't seem that large. Is it really just because using ALL the data to define the correlations is computationally infeasible? Please expand on this point a bit in this section. Or it just doesn't buy you anything? If the latter, then how do you determine how much subsampling is justified before you start to introduce errors?
Equation 1: I just don't get the difference between the Ps and Pt terms. Pt I get. Ps I don't. For instance, in this method, soundings that are 0.5 km from the center of the grid box are 4 times more likely to be selected than soundings 1 km from the center. Even when the spatial resolution of the soundings themselves is 10 km!, and typically decorrelation lengths of CO2 and CH4 in the atmosphere are more like 100+ km! It seems like an exponential structure for Ps makes a lot more sense. Or at least something like $h_s' = max(h_s, h_{min})$ where $h_{min}$ is some minimum resolution distance. (And for Co2 and CH4 I would argue making this at least 10-20 km). There is no physical justification actually cited for these functional forms. If the functional form for Ps is changed to exponential, then obviously the entire discussion from likes 122-134 could be shortened or eliminated.

Line 268: …ecological applications. Please provide some references here.

**Technical Comments**
Line 229: "is a Lagrange multiplier" is missing the actual variable.

Line 316 (and later): ST is never defined. Suggestion you modify the sentence here to say …performance of spatio-temporal (ST) versus…

Page 10, top: I disagree with the conclusions stated here. The MAE and RMSE even for the 7d results seem really only marginally better for ST. And 1d pure spatial, which seems like a more fair comparison as the ST is also done at the daily scale, seems to do as well or better than ST! Also the % lying outside the different uncertainty bounds doesn't seem useful, especially considering that the numbers are significantly less than that expected from pure Gaussian errors. Could the authors explain why they are so much less?

Conclusions near line 404: Again I just don't the ST approach being better. It is only marginally better than 7d, and is slightly worse than 1d. At best this is a wash. Please reword.

**Citations**

Houweling, S., et al, 2015. "An intercomparison of inverse models for estimating sources and sinks of CO2 using GOSAT measurements." *Journal of Geophysical Research: Atmospheres* 120.10, 5253-5266.

Massart, Sébastien, et al., 2016."Ability of the 4-D-Var analysis of the GOSAT BESD XCO 2 retrievals to characterize atmospheric CO 2 at large and synoptic scales." *Atmospheric Chemistry and Physics* 16.3, 1653-1671.

---

## Author Comment (AC1) · 5 Jan 2017

We thank the reviewers for their positive assessment of the manuscript and for their helpful comments. In the text below, we include the reviewer's original comments in italics, while our responses are listed in regular font.

Reviewer: The aims of this study are scattered in the introduction and should be clearly presented in the end of the introduction

Authors: We condensed the goals of the study into one paragraph, now Lines 70-77.

Reviewer: As to spatial-temporal approach, it seems some recent developments since 2011 have been missed, which should be included. Please see reference 1 for details.

Authors: Both provided references are included now (Lines 69 and 325)

[Figure]

Reviewer: Samples size is missing for the three datasets. Please provide.

Authors: The sample was specified in Lines 100.

Reviewer: It is not clear what software was used for this study. Please refer your readers to it so that they could apply your method to their study.

Authors: We specified the software package in Line 77.

Reviewer: The accuracy measures used, MAE and RMSE, are data unit/scale and variation dependent as detailed in reference 2. Please see the recommendations in this reference for accuracy measure selection.

Authors: Per reviewer suggestion, we included two unit-independent error measures, RMAE and RRMSE in the revised version of the manuscript., Lines 326-333 and Table 1.

Reviewer: A statistical test of the cross-validation results in Table 1 may provide more convincing evidence to show the difference between the methods compared.

Authors: Please see our response to Review 2, to a similar comment.

Reviewer: The conclusion: it is largely repeating what has been presented in the previous sections. It could be condensed by removing the repetitions.

Authors: We considerably shortened the second paragraph of the Conclusion section, per reviewer request.

Minor issues:

Reviewer: Spell out GOSAT, IASI and GOME-2 in the abstract or delete them.

Authors: Corrected.

Reviewer: Lines 98-99: this sentence suggests that the method is only applicable for a small region. Please revise and clarify.

[Figure]

Authors: The confusing sentence has been deleted.

Reviewer: Lines 144-145: are 'generalized product-sum model' and 'generalized product-sum covariance model' the same? Please keep the name consistent in the paper.

Authors: Corrected.

Reviewer: Line 173: delete one 'then'.

Authors: Corrected.

Reviewer: For XCO2, only 6 day data were used. Is this too short for ST method? Is it a factor for the poor performance of ST method?

Authors: In the study by Hammerling et al., 2012 the authors examined optimal temporal aggregation time periods for XCO2 retrievals by analyzing the tradeoff between not having too much temporal variability vs. having sufficient observations in the context of spatial-only interpolation approach. They reported that 4-days temporal resolution gave the best results which points out to the fact that expected decorrelation temporal "length" of CO2 field is at the order of magnitude of synoptic scales. Based on their analysis, 6d should not be too short. We believe that the factor affecting "poor" performance of ST (in case of XCO2) compared to what could be expected are different. We changed the following paragraph to make it more clear (Lines 391-399): " The difference between the performance of ST and S-approaches obtained through cross-validation becomes most pronounced in processing datasets with measurement errors that are spatially but not temporally correlated. In these cases, an ST approach can use data from adjacent time periods to create the estimate, data that do not have the same regional, spatially-correlated biases. Although the resulting estimate may appear inferior during cross-validation, this is because that estimate will not reproduce regional biases in data from the time slice of interest." Note that the cross-validation errors and true errors are not identical, the former is just an estimate of the latter. The direct conclusion from this statement is that ST could perform worse in cross-validation, while in fact it filters regionally correlated measurement errors (not reproduced in time) which brings the focus back on whether the leave-one-out cross validation is the best method for validating this and similar techniques, although it has been used in a series of recent papers (Guo et al., 2013; Zeng et al., 2013, Tadic et al., 2015; Zeng et al., 2016). Please see the response to Reviewer 2. We also checked (not shown in the paper) the timeseries of estimates at selected locations where the difference between S and ST was particularly pronounced. We found that S method produced unrealistically high oscillations in estimates along the temporal axis, while ST kept estimated signal much smoother, which also supports the conclusions. A hypothetical alternative approach to improve the apparent cross-validation performance would be to explicitly model the retrieval error covariance matrix, instead of assuming the independence of retrieval errors, or, in other words, to isolate measurement clusters having regionally correlated errors. However, such information is usually not available. Interestingly, the very difference in performance between ST vs. S could be used to address this important, but still not fully resolved issue.

Reference: Hammerling, D. M., A. M. Michalak, and S. R. Kawa (2012), Mapping of $CO_2$ at high spatiotemporal resolution using satellite observations: Global distributions from OCO-2, J. Geophys. Res., 117, D06306, doi:10.1029/2011JD017015.

Reviewer: Lines 345-346: ST method seems not that poor for GOME-2 data. Please revise.

Authors: Corrected.

References:

Guo, L., Lei, L. and Zeng, Z.: Spatiotemporal correlation analysis of satellite-observed $CO_2$: Case studies in China and USA. Geoscience and Remote Sensing Symposium (IGARSS), 2013 IEEE International, 21-26 July, Melbourne, VIC, 2013.

Zeng, Z., LiPing, L., L. LiJie, G., Li, Z., Bing, Z.,: Incorporating temporal variability to improve geostatistical analysis of satellite-observed CO2 in China, Chinese Science Bulletin, 58(16), 1948-1954, 2013.

Zeng, Z.; Lei, L.; Hou, S.; Ru, F.; Guan, X.; Zhang, B., A regional gap-filling method based on spatiotemporal variogram model of columns, IEEE Transactions on Geoscience and Remote Sensing, 2014, 52, 3594-3603.

Guo, L., Lei, L., Zeng, Z.C., Zou, P., Liu, D. and Zhang, B., 2015. Evaluation of spatiotemporal variogram models for mapping Xco 2 using satellite observations: A case study in china. IEEE Journal of Selected Topics in Applied Earth Observations and Remote Sensing, 8(1), pp.376-385. Tadić, J., Qiu, X., Yadav, V. and Michalak, A.: Mapping of satellite Earth observations using moving window block kriging, Geosci. Model Dev., 8, 1–9, doi:10.5194/gmd-8-1-2015, 2015.

Guo, L., Lei, L., Zeng, Z.C., Zou, P., Liu, D. and Zhang, B., 2015. Evaluation of spatiotemporal variogram models for mapping Xco 2 using satellite observations: A case study in china. IEEE Journal of Selected Topics in Applied Earth Observations and Remote Sensing, 8(1), pp.376-385.

Zeng, Z., Lei, L., Strong, K., Jones, D. B. A., Guo, L., Liu, ., Deng, F., Deutscher, N. M., Dubey, M. K., Griffith, D. W. T., Hase, F., Henderson, B., Kivi, R., Lindenmaier, R., Morino, I., Notholt, J., Ohyama,H., Petri, C., Sussmann, R., Velazco, V., A., Wennberg, P., O., and Lin, H.: Global land mapping of satellite-observed CO2 total columns using spatio-temporal geostatistics, International Journal of Digital Earth, DOI: 10.1080/17538947.2016.1156777, 2016.

---

## Author Comment (AC2) · 5 Jan 2017

Reviewer: I will be upfront and say that as reviewer, I am not well- ‐versed in the geostatistical estimation literature, and am rather an expert on these carbon cycle variables themselves. So my review will focus less on the details of this particular approach, and rather some bigger picture questions. My main complaint on this work, which honestly is more a complaint about the entire field who does this, and is not particular to this paper, is that it fails to really explain the utility of kriged satellite data beyond simply "pretty pictures". Most data users who attempt to extract scientific results from the data do not use 3D maps. The reason is the data assimilation systems typically ingest the sounding (level- ‐2) data directly (e.g., Houweling et al., 2016; Massart et al., 2016). Therefore, some commentary (like a paragraph in the introduction section) on the use of level- ‐3 maps vs. direct data assimilation approaches

would be worthwhile, perhaps pointing to scientific results using this method that would have been missed otherwise.

Authors: We would like to emphasize that the methodological advances we presented go beyond the application space defined by three chosen examples. Also, considering the presented method purely as a "mapping" method represents an over-simplification. The method can be, of course, used to produce maps, but it is also capable of upscaling the observations providing estimates at support larger than the support of observations, with associated uncertainties. Example: Imagine that we intend to compare XCO2 derived from OCO-2 and GOSAT retrievals. The direct comparison is not possible because of at least three reasons: (a) the measurements are not collocated (and thus mapping is required), (b) the averaging kernels are different, and (c) the measurements have considerably different spatial statistical properties - support (and thus upscaling of the OCO-2 observations is required). The differences in support can cause substantial differences in reported values (see Tadic and Michalak, 2016). The example shows that even a simple comparison of the same physical quantity measured by two satellites requires a relatively complicated mapping and upscaling methods. The similar conceptual problem remains when model outputs, usually given at regular grids and standardized support, are compared to observational datasets, and when satellite products have to be compared to in situ observations (for example Aircore or aircraft profiles) which are not collocated. Interpolated products could be useful for providing background concentration estimates or initial condition estimates, for example in inverse modeling studies. NOAA has recognized the problem stemming out from the inconsistency in spatio-temporal coverage, and provided justification for mapping: http://www.esrl.noaa.gov/gmd/ccgg/globalview/index.html. The data assimilation systems indeed ingest observations rather than mapped products, but mapping and upscaling method presented here is not limited to greenhouse gas measurements. While transitions of the type Level 2(obs.) - > Level 4(flux patterns), and later eventually Level 4 - > Level 3(maps) are possible, not all the physical quantities have Level 4 data. Actually the solar induced fluorescence (SIF) is a good example. Level 3 data

have been used or generated in a number of recent studies, at the same time providing the insight into their value and scope of application: Liu et al., 2012; Basu et al., 2014; Maksyutov et al., 2013, etc. There are at least few studies we are aware of that currently use mapped and upscaled products: 1) Shiga at al. (Carnegie institution for science) currently use spatio-temporally (ST) mapped SIF as ancillary data in inversion studies, and preliminary results show that ST product is more consistent with atmospheric CO2 observations, then purely spatial product. The publication will follow soon (private communication). 2) Zheng et al. (Yale University) currently use mapped SIF product to study the impact of extreme drought on photosynthesis. The publication will follow soon, too (private communication).

References:

Tadić, J. M., & Michalak, A. M. (2016). On the effect of spatial variability and support on validation of remote sensing observations of CO2. Atmospheric Environment, 132, 309–316.

Liu, J., I. Fung, E. Kalnay, J.-S. Kang, E. T. Olsen, and L. Chen (2012), Simultaneous assimilation of AIRS Xco2 and meteorological observations in a carbon climate model with an ensemble Kalman filter, J. Geophys. Res., 117, D05309,doi:10.1029/2011JD016642.

Basu, S., M. Krol, A. Butz, C. Clerbaux, Y. Sawa, T. Machida, H. Matsueda, C. Frankenberg, O. P. Hasekamp, and I. Aben (2014), The seasonal variation of the CO2 flux over Tropical Asia estimated from GOSAT, CONTRAIL, and IASI, Geophys. Res. Lett., 41, 1809–1815, doi:10.1002/2013GL059105.

Maksyutov, S., Takagi, H., Valsala, V. K., Saito, M., Oda, T., Saeki, T., Belikov, D. A., Saito, R., Ito, A., Yoshida, Y., Morino, I., Uchino, O., Andres, R. J., and Yokota, T.: Regional CO2 flux estimates for 2009–2010 based on GOSAT and ground-based CO2 observations, Atmos. Chem. Phys., 13, 9351–9373, doi:10.5194/acp-13-9351-2013, 2013.

Reviewer: Beyond that, the few basic statistics on the quality of the spatio-temporal (ST) method over and above pure spatial methods do not really argue that the ST approach buys you much. The actual statistics given in Table 1 are really rather similar between pure spatial vs. the ST method. So the paper seems to argue that this is really useful, but the data really don't back it up. My read is that 1-3 day spatial approaches are really quite adequate for this purpose. Finally, the validation approach is probably not valid for the GOSAT case. This is because there are only ∼14 orbits per day, and huge swaths of the globe are missing even if all the data are used. Therefore, you don't really learn the error statistics unless you perform a simulation-based test where you start with a "true" map, sample it like the satellite would, along with realistic observation errors, and then run it through the kriging algorithm to reconstruct the 1-day map. This paper would be much enhanced if such a realistic validation test were performed. I realize the authors can easily say "beyond the scope of this paper" because what I am suggesting is not easy, but it is really the only way I can see to get at the true errors in the proposed algorithm.

Authors: Two comments listed above are related to each other and will be handled together. First, the statistics differs in three test cases so the general conclusions would be pretentious. We provided potential explanations for a poorer performance of the ST approach in GOSAT cross-validation (Lines 384-399). We would like to point out to our reply to Reviewer 1 about errors that are spatially but not temporally correlated, and its effect on the apparent poorer performance of the method, in one satellite case and based on the specific metrics used here. The poorer performance could actually result from ST method providing more accurate, unbiased estimates, yet this has to be further studied. While leave-one-out cross validation might not be the best method for providing the accurate error statistics (as we pointed out both in our reply to reviewer 1 and in the manuscript (Lines 394-396: "Although the resulting estimate may appear inferior during cross-validation, this is because that estimate will not reproduce regional biases in data from the time slice of interest.") it has a long tradition as tool used to assess the performance of similar methods, and we decided to present its results, but pointing

out to potential problems in using it. The synthetic study suggested by Reviewer only for GOSAT case could be usable, but there are at least two entailed problems: (1) we would like to keep consistent error statistics tools across all examples and, (2) synthetic experiment like the suggested one would require a realistic individual retrieval uncertainty estimate. Making assumption about the individual retrieval uncertainty would just mean pushing the problem down the line. There is a long list of studies (see Reference in response to reviewer 1) which all relied upon leave-one-out cross validation done in the manner similar to the one from this study, and to assure comparability between the results we followed the same pattern.

Reviewer: Abstract: Makes that statement that this approach only requires a limited number of assumptions – that "the observable quantity exhibits spatial and temporal correlations that are inferable from the data." But this seems like a single assumption? Are there more assumptions? Please reword as necessary.

Authors: Corrected.

Reviewer: Section 2.1 I don't get why subsampling is necessary. The data volumes don't seem that large. Is it really just because using ALL the data to define the correlations is computationally infeasible? Please expand on this point a bit in this section. Or it just doesn't buy you anything? If the latter, then how do you determine how much subsampling is justified before you start to introduce errors?

Authors: The subsampling is always necessary in moving window approach to preferentially focus on variability near (in spatio-temporal sense) an estimation location, independently of the available number of observations. In addition, in case of GOME-2 and IASI the number of available observations significantly grows if multiple time slices are included. For example, the covariance matrix covering two weeks of IASI data would have 3 billion entries. It is clear that some kind of subsampling has to be done in order to keep the problem at computationally feasible levels. The estimates do not degrade gradually when subsampling fewer and fewer data points, they rather stay fairly

constant over a certain range of subsampled dataset sizes, and then start to degrade at certain level. To determine such a level one has to produce a series of estimates for the same location while subsampling fewer and fewer measurements, until estimates start to differ above the acceptable threshold. We implemented similar approach (Lines 100-101).

Reviewer: Equation 1: I just don't get the difference between the Ps and Pt terms. Pt I get. Ps I don't. For instance, in this method, soundings that are 0.5 km from the center of the grid box are 4 times more likely to be selected than soundings 1 km from the center. Even when the spatial resolution of the soundings themselves is 10 km!, and typically decorrelation lengths of CO2 and CH4 in the atmosphere are more like 100+ km! It seems like an exponential structure for Ps makes a lot more sense. Or at least something like hs' = max(hs, hmin) where hmin is some minimum resolution distance. (And for Co2 and CH4 I would argue making this at least 10-Â∎‐20 km). There is no physical justification actually cited for these functional forms. If the functional form for Ps is changed to exponential, then obviously the entire discussion from likes 122-Â∎‐134 could be shortened or eliminated.

Authors: The choice of the form of the subsampling function is one of the subjective choices the modeler has to make. Instead of arguing why we did select Ps and Pt forms as in the paper, we would like to explain why the 1/ht2 was not used. The satellite data (Level 2) come with continuous spatial and discretized temporal coordinates. Phrased differently, data are temporally pre-aggregated (day 1, day 2, etc.). Any form of the temporal component of the sampling function from 1/htn family would lead to sampling only from the time slice of the estimation location because 1/0 would result in an infinite sampling probability for such observations, unlike observations in other time slices. So the selection of exponential form for the temporal component partially came out of necessity. We do not quite understand the argument about 0.5/1km distance from the center given the spatial resolution 10km (GOSAT). While sampling probability is indeed 4 times higher for 0.5km distant observations, the number of available observations in

combination with selected number of points to be subsampled leads to sampling of all of those points regardless a relative sampling probability difference between them. It is more important that the sampling probability between points 10 and 100 km away differs by factor of 100. We absolutely accept the idea that sampling probability function form can take different shapes, and that it actually can account for anisotropy, and the choice presented in the paper represents just one example. There is indeed no physical justification for the forms selected, like reviewer commented, and in principle it could be replaced with exponential form. However, we do not see that it makes the conceptual presentation of the approach stronger.

Reviewer: Line 268: . . .ecological applications. Please provide some references here.

Authors: We included two references in the Line 269 per reviewer suggestion.

Reviewer: Line 229: "is a Lagrange multiplier" is missing the actual variable.

Authors: Corrected.

Reviewer: Line 316 (and later): ST is never defined. Suggestion you modify the sentence here to say . . .performance of spatio-Â■‐temporal (ST) versus. . .

Authors: Corrected. Thank you for the suggestion.

Reviewer: Page 10, top: I disagree with the conclusions stated here. The MAE and RMSE even for the 7d results seem really only marginally better for ST. And 1d pure spatial, which seems like a more fair comparison as the ST is also done at the daily scale, seems to do as well or better than ST! Also the % lying outside the different uncertainy bounds doesn't seem useful, especially considering that the numbers are significantly less than that expected from pure Gaussian errors. Could the authors explain why they are so much less?

Authors: It is questionable if a comparison between 1d spatial and ST is more fair. In the case of a comparison between ST and 7d spatial we actually produce estimates using the same data, it is just that in the ST approach the temporal covariance between them is properly characterized, and in the 7d spatial it is not the case. But one might argue that it is more fair, given that we use the same observational data to produce estimates. The 1d spatial case can produce apparently better statistics because of the biases that are spatially correlated but do not reproduce in time, like we mentioned before, and thus it comes back to the question of the selection of the best error metrics, because leave-one-out cross validation yields the numbers which show the degree of regional consistency between the data, not its true accuracy. This discussion is provided in the paper at Lines 391-399. Yet unpublished results show that, based on BIC score, ST method yields SIF estimates that are more consistent with atmospheric observations of $CO_2$ (private communication, Shiga et al., Carnegie Institution for Science).

Reviewer: Conclusions near line 404: Again I just don't the ST approach being better. It is only marginally better than 7d, and is slightly worse than 1d. At best this is a wash. Please reword.

Authors: We cite the commented sentence: "The method generally yields more precise and accurate (and unbiased) estimates compared to spatial method which used the same observations but assumed perfect temporal correlation between data." We believe it is clear that this sentence was meant to express that ST yield better results than S 7d ("...compared to spatial method which used the same observations but assumed perfect temporal correlation between data..."). It did not mean to address 1d spatial vs ST comparison. In case of GOSAT, IASI and GOME-2 ST yielded 6, 9 and 4% lower MAE. Those values are consistent with other studies that evaluated ST vs spatial (Guo et al., 2013, Zeng et al., 2013 and 2016).

At the end, the reported statistics for GOME-2 is now slightly changed in the Table 1, as we found a small glitch in the code we used to process GOME-2 dataset. Now, the S method was found to produce better estimates than ST approach only in GOSAT 1d case, for the reason we discussed above. In all other cases, ST method was found to yield best error statistics.

---

## Author Comment (AC3) · 5 Jan 2017

In response to the requests by Executive Editor we added the Code and/or data availability section to the manuscript. However, we believe that providing a version number in the title of the paper would be unnecessary. We present a conceptual statistical modeling approach, and nature of this paper is similar to the nature of several other papers published at GMD without providing version number in the title. For example:

A sparse reconstruction method for the estimation of multi-resolution emission fields via atmospheric inversion, J. Ray, J. Lee, V. Yadav, S. Lefantzi, A. M. Michalak, and B. van Bloemen Waanders

Simulation of trace gases and aerosols over the Indian domain: evaluation of the WRF-Chem model, M. Michael, A. Yadav, S. N. Tripathi, V. P. Kanawade, A. Gaur, P. Sadavarte, and C. Venkataraman

Mapping of satellite Earth observations using moving window block kriging J. M. Tadić et al.

---

## Editor Decision (ED1)

Dear Jovan Tadic,

we do not invent rules, just because we like rules. The meaningful questions behind the rule of presenting version numbers in the title is, whether the method you are publishing here, will be modified in future, e.g., by additionaly features or by bug fixes. If so, it will become important in the future to know, which version was used for the production of published results. Therefore we require version numbers in the titles of the articles.

The fact, that other articles in the past did not provide this version number is no argument at all, because

1. the three articles have been published before our Editorial 1.1 was published. Thus at that time is was not an established rule that version numbers must be given.
2. Even if the articles would have been published afterwards, the fact that our enforcing failed in the past is not a valid argument to prevent following this rule in the presence.

Best regards,
 Astrid Kerkweg (Executive Editor)

---

## Author Response (AR2)

**Reply to Editor:**

We added the version number to the title, as well as explained license terms in the Section 6 ("Code availability").

Authors